# The Place of Botulinum Toxin in Spastic Hemiplegic Shoulder Pain after Stroke: A Scoping Review

**DOI:** 10.3390/ijerph20042797

**Published:** 2023-02-04

**Authors:** Pieter Struyf, Lisa Tedesco Triccas, Fabienne Schillebeeckx, Filip Struyf

**Affiliations:** 1Department of Rehabilitation Sciences and Physiotherapy, Faculty of Medicine and Health Sciences, University of Antwerp, B-2610 Wilrijk, Belgium; 2Department of Rehabilitation Medicine, Research School Caphri, Maastricht University Medical Center, 6200 MD Maastricht, The Netherlands; 3Adelante Zorggroep, 6229 HX Maastricht, The Netherlands; 4REVAL, Faculty of Rehabilitation Sciences, Universiteit Hasselt, B-3590 Diepenbeek, Belgium; 5Department of Clinical and Movement Neuroscience, Institute of Neurology, University College London, London WC1N 3BG, UK; 6Department of Physical Medicine and Rehabilitation, University Hospital Leuven, B-3000 Leuven, Belgium

**Keywords:** spastic hemiplegia, botulinum toxin, shoulder pain

## Abstract

Stroke is a common pathology worldwide, with an age-standardized global rate of new strokes of 150.5 per 100,000 population in 2017. Stroke causes upper motor neuron impairment leading to a spectrum of muscle weakness around the shoulder joint, changes in muscle tone, and subsequent soft tissue changes. Hemiplegic shoulder pain (HSP) is the most common pain condition in stroke patients and one of the four most common medical complications after stroke. The importance of the appropriate positioning and handling of the hemiplegic shoulder for prevention of HSP is therefore of high clinical relevance. Nevertheless, HSP remains a frequent and disabling problem after stroke, with a 1-year prevalence rate up to 39%. Furthermore, the severity of the motor impairment is one of the most important identified risk factors for HSP in literature. Spasticity is one of these motor impairments that is likely to be modifiable. After ruling out or treating other shoulder pathologies, spasticity must be assessed and treated because it could lead to a cascade of unwanted complications, including spastic HSP. In clinical practice, Botulinum toxin A (BTA) is regarded as the first-choice treatment of focal spasticity in the upper limb, as it gives the opportunity to target specifically selected muscles. It thereby provides the possibility of a unique patient tailored focal and reversible treatment for post stroke spasticity. This scoping review aims to summarize the current evidence of BTA treatment for spastic HSP. First, the clinical manifestation and outcome measures of spastic HSP will be addressed, and second the current evidence of BTA treatment of spastic HSP will be reviewed. We also go in-depth into the elements of BTA application that may optimize the therapeutic effect of BTA. Finally, future considerations for the use of BTA for spastic HSP in clinical practice and research settings will be discussed.

## 1. Introduction

Stroke is a common pathology worldwide, with an age-standardized global rate of new strokes of 150.5 per 100,000 population in 2017 [1]. It is reported as the third leading cause of disability. Stroke causes upper motor neuron impairment leading to a spectrum of muscle weakness around the shoulder joint, changes in muscle tone, and subsequent soft tissue changes. Hemiplegic shoulder pain (HSP) is the most common pain condition in stroke patients and one of the four most common medical complications after stroke, together with depression, falls, and urinary tract infections [2,3]. The importance of the appropriate positioning and handling of the hemiplegic shoulder for prevention of HSP is therefore of high clinical relevance [4]. Nevertheless, HSP remains a frequent and disabling problem after stroke, with a 1-year prevalence rate up to 39% [5]. In addition, HSP reduces participation and worsens outcomes in rehabilitation [6] and has a very negative impact on the quality of life in stroke survivors [7]. The most often identified predictors of HSP in stroke are a relatively younger age (less than 70 years), female gender, sensory impairment, left-sided hemiparesis, hemorrhagic stroke, hemispatial neglect, positive past medical history, and a higher than medium severity of disability (>14/42) on the National Institutes of Health Stroke Scale score) and spasticity [5] (Table 1). Furthermore, the severity of the motor impairment is one of the most important identified risk factors for HSP in literature [8,9,10,11,12]. Most of these predictive factors are unfortunately not modifiable. Spasticity on the contrary is likely to be modifiable, but it is not the sole cause of HSP. Independent of spasticity, other pathologies (e.g., frozen shoulder, rotator cuff tendinopathy), sympathetic dysregulation (e.g., shoulder hand syndrome), cervical involvement, and central post stroke pain could be present [13]. After ruling out or treating these pathologies, spasticity must be assessed and treated because it could lead to a cascade of unwanted complications, including spastic HSP [14].

In clinical practice, Botulinum toxin A (BTA) is regarded as the first-choice treatment of focal spasticity in the upper limb, as it gives the opportunity to target specifically selected muscles. It thereby provides the possibility of a unique patient tailored focal and reversible treatment for post stroke spasticity [15,16]. Conversely, in clinical practice the use of BTA in spastic HSP is very variable. A survey of clinical practice among physicians and physiotherapists with a special interest in spasticity management, showed that 86.8% would consider injecting BTA for HSP associated with spasticity but only 54.4% agreed or strongly agreed that BTA is an effective treatment for HSP. Only 8.8 % would choose BTA as a first line treatment [17]. It should be stated that BTA in HSP is not a stand-alone therapy. Although they are not the scope of this article, adjunct therapies, such as stretching and active or passive physiotherapy techniques, may improve the treatment results [18].

This scoping review aims to summarize the current evidence of BTA treatment for spastic HSP. First the clinical manifestation and outcome measures of spastic HSP will be addressed, and second the current evidence of BTA treatment of spastic HSP will be reviewed. We also go in-depth into the elements of BTA application that may optimize the therapeutic effect of BTA. Finally, future considerations for the use of BTA for spastic HSP in clinical practice and research settings will be discussed.

## 2. Spastic Hemiplegic Shoulder Pain

### 2.1. What Are the Clinical Manifestations of Spastic Hemiplegic Shoulder Pain?

Approximately 38% of stroke survivors develop spasticity over time [19]. This can occur at any time but is usually seen between 1 and 6 weeks after a stroke [19]. Spasticity is traditionally defined as a motor disorder characterized by a velocity-dependent increase in muscle tone, caused by the increased excitability of the muscle stretch reflexes [20]. To do more justice to the array of other spasticity related phenomena, the definition was further elaborated to ‘a disturbed sensory-motor regulation following an upper motor neuron lesion, manifesting as an intermittent or continuous involuntary muscle activity’ [21]. Symptoms may include hypertonicity (increased muscle tone), clonus (a series of rapid muscle contractions) exaggerated deep tendon reflexes, muscle spasms (disinhibited normal flexor withdrawal reflexes), spastic dystonia (tonic muscle contraction in absence of a phasic stretch or volitional command), spastic co-contraction (inappropriate recruitment of antagonist muscles), or a mixture of these elements [16]. Pain may subsequently occur. Due to the wide variety of possible manifestations of spasticity, its assessment becomes very complex.

Together with spasticity, the loss of muscle strength and motor control will undoubtedly further alter the kinematics around the shoulder complex in stroke patients [17]. With time, these kinematic alterations of the shoulder joint result in biomechanical changes in muscles and soft tissues, such as changed properties of the muscle contractile elements and soft tissue, resulting in changed elasticity and viscosity, which contribute to the velocity-dependent resistance to stretching. This is the so called non-neural aspect of spasticity.

These soft tissue changes can evolve to the formation of static contractures [22]. Contractures are characterized by the combination of increased stiffness and loss of range of movement at a joint. Histologically these changes are associated with relatively shorter muscle fascicles (compared to matched controls) and diminished serial sarcomeres [23,24]. These changes can lead to a vicious circle of further muscle shortening leading to a further increase of spasticity, which can result in more pain. In hemiparetic patients (having a variable level of preserved motor control) contractures further interfere with volitional movement of the upper limb. In hemiplegic patients (with the absence of any volitional shoulder movement), shoulder adduction contractions lead to inadequate hygiene and maceration of the skin in the axilla region, and pain when caregivers mobilize the shoulder for dressing and hygiene purposes.

In general, it is considered that spasticity involves mainly anti-gravity muscles, resulting in an extension pattern in the lower limb and flexion pattern in the upper limb [25]. The anti-gravity concept is not entirely applicable to the shoulder girdle, because spasticity of adductors, flexors, and internal rotators is most often observed [2]. A possible explanation is the location of the brain lesion (mainly severity of a lesion in the inhibitory corticospinal and dorsal reticulospinal tract) [26]. Other contributors are the adductor muscles, which are stronger than the abductors at the shoulder level. Also, at the elbow level, the flexors are stronger than the extensors [27].

Based on a differentiated posture and arm movement analysis, five characteristic arm spasticity patterns (ASP I-V) were defined by Hefter et al. [28] with respect to the position of the shoulder, elbow, forearm, and wrist joints. These patterns were verified using data from a worldwide noninterventional Upper Limb International Survey. By clinical observation, spastic arm postures in 94% of 665 poststroke patients could be assigned to one of these five ASPs. The most frequent pattern of arm spasticity was ASP III (41.8%) with internal rotation and adduction of the shoulder and flexion at the elbow coupled with a neutral positioning of the forearm and wrist. Nearly all patterns show an adducted and internally rotated shoulder, with flexion at the elbow. Only pattern V shows shoulder adduction, internal rotation but elbow extension, accounting for 9.8% of cases [29]. Moreover, the more powerful internal rotators (compared to the external rotators) of the shoulder share the same insertion as the adductors and are mainly the same muscles. Therefore, a spastic pattern with an adducted shoulder also involves an internally rotated shoulder [30]. Occasionally, a pattern showing shoulder abduction and internal rotation is observed in clinical practice but not described in any of Hefter’s categories [30].

The painful spastic hemiplegic shoulder is generally presented in an adducted and internally rotated position. Spasticity, loss of muscle strength and motor control, alters the kinematics of the shoulder joint. Beside the wide variety of possible presentations of the neural aspect of spasticity, there is also a non-neural aspect that can lead to static contractures, evolving to a vicious cycle of further muscle shortening, spasticity, and more pain. However, measurement of spasticity is also an important issue to be discussed.

### 2.2. What Are Adequate Outcome Measures of Shoulder Pain and Spasticity of the Shoulder Girdle?

Shoulder pain is easily assessed by the widely used Numeric Rating Scale (NRS) or visual analogue scale (VAS). The latter is more suited for patients suffering from dysphasia. In addition, the Shoulder Pain and Disability Index (SPADI) can be used to specifically target the shoulder region. Both the NRS and SPADI are deemed psychometrically reliable and valid in patients with shoulder pain [31,32,33,34].

No specific outcome measures are found in the literature concerning spasticity of the shoulder girdle. Looking at outcome measures for spasticity in general, the Modified Ashworth Scale (MAS) is the most used outcome measure at body function level. However, this scale has been criticized regarding its measurement properties (construct validity and sensitivity) [35]. The MAS measures changes in resistance to passive movement on an ordinal scale, which is only one spasticity related phenomenon. Other phenomena, such as hypertonicity, clonus, exaggerated deep tendon reflexes, muscle spasms, spastic dystonia, and spastic co-contraction are not measured by the MAS [36]. The MAS is therefore a poor surrogate marker for spasticity, only measuring it indirectly. Consequently, one could contest the relevance of the MAS as the most frequently used tool in clinical trials to investigate the effect of a BTA intervention. Other scales used in stroke are the Modified Tardieu, the Disability Assessment, the Fugl–Meyer, the Motor Assessment scale, tone assessment scale, and King’s hypertonicity scale. All these scales, however, also lack reliability and reproducibility [37,38]. In addition, they each focus on only one or a few spasticity related phenomena, not considering the broad picture of spasticity. They are also unable to identify the neural and non-neural contributions to spasticity. Consequently, there is a great need for appropriate scales, that are sensitive enough to pick up subtle changes in the spasticity related phenomena and have a better reliability.

Other than the previously mentioned assessment scales, there are promising techniques to be used in future research, such as isokinetic robotic devices [39,40], surface electromyography (sEMG) [41], measurement and F/M wave ratio in electrophysiology, and ultrasound and shear wave elastography of the spastic muscles and tendons [24].

Isokinetic robotic devices have a good inter-rater reliability, using a standardized range of isokinetic velocities and measurement of peak resistance and range of motion [40]. Combining the robotic device with surface EMG helps to differentiate between neural and non-neural contributors to spasticity [39]. In fact, a randomized clinical trial (RCT) by Lindsay et al. [42], found a significant mismatch between the Tardieu scale and surface EMG activity (a more direct neurophysiological measure of spasticity). The latter turned out to be almost twice as sensitive.

Ultrasound (US) and shear wave elastography provide extra information about the non-neural contributors to spasticity. Changes in morphological muscle and tendon properties related to spastic hemiparesis after stroke can be directly studied at the macroscopic level. US can identify changes in muscle and tendon properties in spastic hemiparesis after stroke, notably reduced muscle thickness and fascicle length [43]. In addition, shear wave elastography can be used to evaluate muscle stiffness in spastic muscles. However, further research is needed to clarify its relationship with the neural components of spasticity and other symptoms of the upper motor neuron syndrome, such as muscle weakness or the occurrence of compensation strategies of the patient. Nevertheless, these techniques hold promise for more objective assessments by US in the diagnosis and follow-up of spastic hemiparesis after stroke [43].

Although there are some promising techniques, especially in the upper limb applicable isokinetic robotic devices, there are currently no spasticity scales available that are sensitive enough to measure subtle changes in the wide variety of spasticity related phenomena of the shoulder girdle. Furthermore, the shoulder girdle itself poses a challenge in identifying which muscles are involved in spasticity.

### 2.3. How Do We Select the Right Muscle for Injection?

To provide an appropriate treatment for spasticity related shoulder pain, a comprehensive assessment is necessary to determine the exact muscles involved.

Knowing the dominant pattern of spasticity in the shoulder helps in defining the muscles that need to be targeted. According to the previous findings on Hefter’s categories [28], the following muscles can be identified as causing adduction and internal rotation; m. pectoralis major, m. latissimus dorsi, m. teres major, and m. subscapularis [44].

There is, however, a lack of evidence about which of the identified muscles are the most important to inject [44]. In clinical practice, choices must be made as to which muscles to target within the boundaries of the total dose allowed for the injection session. Most of the time, evaluation is performed by eliciting resistance to movement at rest and observing of patterns of tightness as the limb is used functionally [2].

As mentioned previously, assessment procedures are often nonsensitive, which makes the injection parameters used in clinical practice heterogeneous [45]. Rather than a patient’s characteristics, clinicians’ beliefs and muscle accessibility are often the dominant factors driving the clinical BTA strategy for post stroke upper limb spasticity [45]. Moreover, when the single goal of pain reduction in spastic HSP is pursued, the same heterogeneity is found concerning muscle selection, dosing, and volumes [46]. In addition, according to this survey, the m. pectoralis major is the most often injected muscle, whereas both the m. subscapularis and m. pectoralis major are equivalently studied in the literature [46]. Therefore, injection of m. subscapularis should be given much more attention in clinical BTA injection practice [47,48,49].

When the selection of muscles for BTA injection in clinical practice remains inconclusive, a useful adjunct could be a selective motor nerve block (SMNB) [50]. SMNBs are easy to perform, innocuous, and the effect only lasts a few hours. The method involves local anesthesia of a nerve, preventing its conduction and causing an immediate decrease in spasticity and voluntary muscle activation. It is then possible to evaluate the passive range of motion and to differentiate between spasticity and contracture, as well as to evaluate the strength of antagonist muscles. Changes in posture and movement can be evaluated, and the involvement of the temporarily inhibited muscle determined. A successful example of this technique is demonstrated in the study by Genet et al. [51], where successive motor nerve blocks were applied to identify the muscles causing an elbow flexor pattern. They showed that, despite the often-injected superficial m. biceps brachii and m. brachioradialis, the muscle that limits elbow flexion the most is m. brachialis. Again, the muscles targeted to reduce involuntary elbow flexion are in clinical practice often chosen based on the ease with which the muscle can be accessed. However, when it comes to the shoulder girdle, there are no studies regarding SMNBs available in the literature which can help in differentiating the different muscles contributing to the spasticity pattern, presumably because of the difficulty in accessing the nerves involved, and their mixed sensorimotor nature. An alternative could be the use of muscle blocks with local anesthetics.

However, there is again a lack of evidence about which of the identified muscles [44] are the most important to inject. One should be aware of the pitfall of only injecting the most easily accessible muscle. A muscle block with local anesthetics could help in the decision-making process.

A pragmatic approach for the early injector could be found in the national guidelines of the royal college of physicians and ‘Guidance for Early Injectors from a Delphi Panel Process’ [43]. The latter recommends, as an expert opinion, the injection of m. Pectoralis and m. Latissimus dorsi. The injection of m. Teres major and m. Subscapularis is only advised for the experienced injector. An additional assessment with a needle EMG is advised before injecting m. teres major.

## 3. BTA

### 3.1. What Is the Place of BTA in Spasticity Treatment?

Current common practice for spasticity treatment includes physical therapy (mobilization and stretching), physical modalities (splinting and casting), oral systemic medications, local interventions (botulinum toxin injections, phenol neurolysis), intrathecal baclofen therapy, and neuro-orthopedic surgical corrections [22]. Several factors influence the treatment options, including severity of spasticity, involvement of muscle groups (location), stage of recovery (acute versus chronic phase), and medical condition of the patient, and should be individualized for best clinical outcomes [16]. Treatment options are clinically categorized as systemic versus focal, and reversible (non-surgical) versus irreversible (surgical). In clinical practice, Botulinum toxin A (BTA) is regarded as the first-choice treatment of focal spasticity in the upper limb. The working mechanism of BTA consists of selective inactivation of peripheral cholinergic neurons by blocking the release of acetylcholine at the neuromuscular junction, causing muscle weakness. BTA produces a temporary and reversible blockade of cholinergic transmission, reaching a peak at 5–10 weeks after injection [52,53]. It gives the opportunity to target specifically selected muscles and thereby provides the possibility of a unique patient tailored focal and reversible treatment for post stroke spasticity [4,15]. Specific guidelines about BTA interventions are available in the national guidelines of the royal college of physicians.

### 3.2. Effectiveness of BTA in Spastic HSP

A systematic review and meta-analysis by Andringa et al. [15] demonstrated robust evidence for BTA being a safe and effective treatment after stroke in reducing resistance to passive movement and improving self-care ability of the affected wrist and fingers. Its therapeutic effect is maintained after repeated treatment cycles [54]. All six RCT’s concerning spasticity-related pain in the shoulder in the review, however, showed nonsignificant Standard effect sizes (SESs). Difficulties of injecting the correct muscle and measurement of resistance to passive movement in the shoulder region, and interference with frequently reported pre-existing shoulder pain are given as possible explanations for these findings [15]. Furthermore, difficulties of injecting the correct muscle are most likely, knowing that none of the included studies used US guided injections. A sensitivity analysis of this review however, found that the spasticity related pain score at follow up showed a significantly larger effect size for those who received BTA earlier after stroke [15]. In the next section, on BTA application, we will go further into the factors that can influence the effectiveness of BTA application in spastic HSP.

Pain reduction due to a direct analgetic effect of BTA has also been suggested. However, the lack of consistent effects and paucity of high-quality studies mean that we have chosen to focus on the anti-spasticity properties of BTA [55]. Two promising randomized, double blind, controlled studies hypothesized that HSP associated with range-limiting spasticity around the shoulder girdle may improve with BTA injections [44,56]. The randomized, double blind, controlled studies by De Boer et al. [57] and Kong et al. [58] however, showed conflicting evidence. This could partially be explained by the small sample sizes of the studies and the inclusion of patients with other confounding pathologies [23,59].

Unfortunately, conducting a meta-analysis on these studies is not feasible, because all studies injected BTA in different combinations of muscles, making them not comparable. Despite some available evidence, and the vast theoretical advantages of BTA on post stroke spasticity, the place of BTA for use in spasticity-related shoulder pain remains far from clear.

### 3.3. BTA Application

#### 3.3.1. Injection Technique

Most studies on the use of BTA for HSP discussed before either do not report the technique they used, or they only used needle EMG to target the muscles involved [44,57,58,60]. This might be a part of the explanation as to why conflicting evidence is available on the effect of BTA for HSP. However, there is strong evidence in favor of the use of an instrumented injection-guiding technique compared with manual needle placement [61]. In our opinion, US guiding is indispensable in the shoulder region. US allows real-time accurate scanning of the targeted muscle, adjacent structures, and needle advancement into the tissue, thus facilitating accurate depth control for needle placement [62].

ES and needle EMG targeting injections carry the risk of injecting the wrong muscle, or even unintended injuries of vascular structures or a lung (pneumothorax), especially in the deep muscles of the shoulder girdle (e.g., m. Subscapularis, m. Teres major).

The use of US can prevent these complications. In addition, as already mentioned before, the efficacy may be maximized by injecting the toxin as close as possible to the motor endplate of the muscle [63,64]. One recent study, by Tan et al. [65], did confirm that a US-guided lateral approach of m. subscapularis for BTA injection is a safe, reliable, effective, and precise treatment that relieves pain, reduces spasticity, and improves shoulder ROM and upper extremity function in hemiplegic patients with HSP. We conclude that the use of US is indispensable in accurately targeting the selected muscle with a BTA injection in the shoulder region.

#### 3.3.2. Dose, Dilution, Number of Injections, Interjection Intervals and Adverse Effects

Another explanation for the conflicting evidence of the effect of BTA for HSP is the wide variety of dose, dilution, and number of injections applied in the research. There are several available dosing and dilution protocols, some of them providing recommendations on number of injections, such as the previously mentioned national guidelines of the royal college of physicians and Guidance for Early Injectors from a Delphi Panel Process’ [43]. Unfortunately, they are all based on expert opinion and differ from one another.

First, doses are not interchangeable between different BTA types. Standard doses are generated for use in clinical practice which are generally well-tolerated, and which work for most patients. Second, the dosing of BTA is often influenced by pragmatic choices that must be made in spreading the total BTA dose over multiple body regions, including the lower limb [30].

In general, it is assumed that a lower concentration and higher volume of administered BTA results in greater diffusion and a larger affected area, but also a greater risk of the spread of BTA into unwanted muscles. Unfortunately, these assumptions are merely based on expert consensus.

A study by Gracies et al. [64] showed that high volume or endplate-targeted BTX-A injections of m. biceps brachii achieved greater neuromuscular blockade, co-contraction and spasticity reduction, and active range of elbow extension improvement, than low volume, nontargeted injections. Applied to the shoulder complex, these findings favor a low volume and end-plate targeted technique.

In general, inter-injection intervals are recommended to be not less than 12 weeks because of a possible immune response, although recent studies indicate that shorter intervals may be applied without toxicological or immunological complications [66,67].

Adverse effects of BTA, although rare and most of the time mild, should also be acknowledged. In the short term they include temporary local pain at the site of injection and diffusion of the toxin into neighboring muscles, causing undesirable weakness. In the long-term, muscle denervation, atrophy, and very rarely, immune resistance can occur [68,69]. Apart from the adverse effects mentioned above, a rare but serious adverse effect is the development of swallowing problems, which occurs more often in proximal upper limb applications of BTA, especially in patients with clinical or subclinical compromised swallowing [68]. Regardless of all these variables, it is, in our opinion, above all important to document per injection session in a structured and standardized format in the patient record about the type of BTA, total dose, dilution, dose per muscle, and number of injections per muscle that is used. In that way, proper evaluations and adjustments can be made.

#### 3.3.3. Timing of Injection

As already mentioned, spasticity can arise soon after a stroke [19]. In a post-stroke spasticity (PSS) cohort, 87 out of 100 hemiplegic patients developed spasticity within 6 weeks, as measured using muscle activation recordings [70]. To implement structural early interventions with BTA, a good prediction of patients at risk of developing spasticity is crucial. A recent review by Tedesco Triccas et al. [71] identified age (≥65 years), motor and somatosensory deficits, and hemorrhagic stroke as the most important predictors for upper limb spasticity within the first month after a stroke.

In current practice, treatment with BTA is usually started in the chronic phase after stroke. This is a phase where secondary complications already have developed [72]. Possible explanations are the variable presence of PSS during the first year after onset, published studies which merely report on treatments during the chronic phase, and reimbursement modalities in specific countries allowing BTA reimbursement in the chronic phase [14]. As spasticity can already arise during the first six weeks following a central nervous system lesion, the early initiation of therapy may be beneficial [73]. Furthermore, contractures can already establish within four weeks post stroke [70]. 52% of stroke survivors developed a contracture within six months [74]. An early BTA intervention could therefore reduce further spasticity development, preventing contractures and deformities that may evolve over the course of the disease. A longitudinal cohort study by Picelli et al. [75] recommends a BTA administration within three months to obtain a greater reduction in muscle tone at 1 and 3 months afterwards, although further research is needed to confirm these findings and study the long-term effects.

A second argument in favor of early BTA intervention is that rehabilitation therapies (such as physiotherapy, occupational therapy) are facilitated when specific learning is activated by activation of specific pathways and depression of unwanted (spastic) pathways by BTA [76]. Furthermore, unlike systemic spasmolytic medication, BTA will not depress the central nervous system and therefore not interfere with neural recovery.

To optimize the treatment effect and prevent the development of contractures, early identification, and treatment of spastic HSP is advised. For early detection we recommend weekly evaluation of spasticity in patients at risk for development of spasticity (Table 2) at least the first six weeks post stroke.

## 4. Discussion

We summarized the predictive factors for the development of HSP and spasticity (Table 1 and Table 2). Unfortunately, an algorithm to identify patients who need rigorous early follow up for the development of HSP and spasticity using predictive factors cannot be compiled based on the current literature. Future studies developing prognostic predictive models are therefore needed.

Spasticity research in general suffers from a lack of standardization. This is due to several factors. First, there is the lack of suitable assessment scales. Although there are some promising outcome measures, none of them are currently used for assessment of spasticity in the shoulder girdle. Secondly, there is no standard regime for the optimal dosage, volume (dilution with physiological saline solution), number of injections per muscle, and injection interval [23,59]. Thirdly, there is a variation of available injection techniques, and an insufficiently clarified role of adjuvant therapies. Many questions arise concerning the most beneficial timing of the first administration. In addition, most trials show difficulties with their methodological quality and have insufficient scientific rigor, in which the role of the industry is often unclear [59].

The need remains for large, high quality, randomized controlled trials (RCT’s) on reliable and valid outcome measures for spasticity, which can measure subtle changes of spasticity and can differentiate between neural and non-neural factors. RCT’s are needed on the effectiveness of BTA in HSP, using BTA in the optimal way by screening, assessing, and treating spastic HSP in an early-stage post stroke patient to perform an early intervention with BTA if needed, using US to target the involved muscles. These steps need to be taken before the intervention can be recommended for routine clinical practice.

There could be a concern about selection bias because of the narrative character of this review. However, while a systematic review methodology was originally planned, an initial systematic search identified the non-standardized, diverse nature of the current BTA related studies, and the small body of available relevant work made a systematic approach and meta-analysis of the data not feasible. Also, the narrative approach provided us with a more nuanced and clinically applicable review to extend knowledge and set research priorities in the area.

## 5. Conclusions

Spastic HSP is a common and disabling condition after stroke. Its assessment and treatment remain challenging. Even though BTA is the first-choice treatment for focal spasticity post stroke, the place of BTA for use in spastic HSP is far from clear. Further research is needed before BTA for spastic HSP can be recommended in routine clinical practice. When BTA treatment is considered for a patient in clinical practice, the application of a structural assessment of spasticity, to make early treatment possible, using US is recommended. In muscle selection, not only the superficial m. pectoralis major and m. latissimus dorsi, but also the deeper m. teres major and m. subscapularis, must be considered. The properties of the injections should be documented systematically in the patient record.

Future research is needed, with large, high quality, randomized controlled trials (RCT’s) on sensitive outcome measures for spasticity and on the effectiveness of BTA in HSP, using BTA in the most optimal conditions.

## Figures and Tables

**Table 1 ijerph-20-02797-t001:** Predictors for the development of hemiplegic shoulder pain.

Age ≤ 65–70 years
Female
Positive past medical history of shoulder pain
Hemorrhagic stroke
Left sided hemiparesis
Hemi-spatial neglect
score NIHSS > 14/42
Severe motor impairment
Spasticity
Somatosensory deficit

**Table 2 ijerph-20-02797-t002:** Advised approach of BTA intervention in spastic HSP.

- Use a structured and standardized format in the patient record specifying: ◦ Type of BTA ◦ Total dose ◦ Dilution ◦ Dose per muscle ◦ Number of injections per muscle - Early intervention - US guided - In case of inconclusive muscle selection: consider muscle block - Muscles to consider: m. pectoralis major m. latissimus dorsi m. teres major m. subscapularis - In case of inconclusive muscle selection: consider muscle block

## Data Availability

Data sharing not applicable. No new data were created or analyzed in this study.

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
