# Peer review of "The Place of Botulinum Toxin in Spastic Hemiplegic Shoulder Pain after Stroke: A Scoping Review"

_ijerph, 2023, doi:10.3390/ijerph20042797_

Round 1
Reviewer 1 Report (Previous Reviewer 2)
Thank you for resubmitting the article for review. The authors have mostly responded to my comments in an acceptable manner. I have no substantive comments, I have one editorial comment. I note one oversight that should be corrected before publication:
''L476: Author Contributions: All authors have read and agreed to the published version of the manuscript'' - Authors' participation should be selected and described according to the template:
'' "Conceptualization, X.X. and Y.Y.; Methodology, X.X.; Software, X.X.; Validation, X.X., Y.Y. and Z.Z.; Formal Analysis, X.X.; Investigation, X.X.; Resources, X.X.; Data Curation, X.X.; Writing – Original Draft Preparation, X.X.; Writing – Review & Editing, X.X.; Visualization, X.X.; Supervision, X.X.; Project Administration, X.X.; Funding Acquisition, Y.Y.”,''
Author Response
We thank the Reviewer for this editorial suggestion. We have added the authors' contributions.
This manuscript is a resubmission of an earlier submission. The following is a list of the peer review reports and author responses from that submission.
Round 1
Reviewer 1 Report
This is a well-written and well-illustrated manuscript reviewing the current evidence of botulinum toxin A (BTA) treatment for spastic hemiplegic spastic shoulder (HSP). For this, the authors first addressed the clinical manifestations and outcome measures of spastic HSP and second reviewed the current evidence of BTA treatment of spastic HSP. They finally detailed the elements of BTA application suggesting an approach aiming to optimize the therapeutic effect of BTA and discussed future considerations for the use of BTA for spastic HSP in clinical practice and research setting. Overall, the report is well organized. Yet, some changes should be reconsidered.
Minor revisions:
· Abstract:
· Appropriatepositioningè appropriate positioning
· In clinical practice Botulinum toxin A (BTA) is regarded asè In clinical practice, Botulinum toxin A (BTA) is regarded as
· Introduction:
· It is reported as the third most leading cause of disability (29)è It is reported as the third most leading cause of disability
· Is the reference 2 (Marciniak C. Poststroke hypertonicity: upper limb assessment and treatment. Top Stroke Rehabil 2011; 18: 179-194. DOI: 10.1310/tsr1803-179) after the sentence “Hemiplegic shoulder pain (HSP) is the most common pain condition in stroke patients and one of the four most common medical complications after stroke, together with depression, falls, and urinary tract infections”, concordant? (the paper doesn’t contain this information), verify please.
· 2.1. What is the Clinical manifestation of spastic hemiplegic shoulder pain?
o What is the Clinical manifestation of spastic hemiplegic shoulder pain? è What are the clinical manifestations of spastic hemiplegic shoulder pain?
o Due to the wide variety of possible manifestations of spasticity its assessment becomes very complex. è Due to the wide variety of possible manifestations of spasticity, its assessment becomes very complex.
· 2.2. What are adequate outcome measures of spasticity?
o The authors should emphasize the outcome measure of “shoulder” spasticity. Are there any with particular interest for “shoulder”.
o The authors should also discuss adequate outcome measure of ”pain”, as they briefly discuss later the possible effect of botulinum toxin on pain.
· 2.3. How do we select the right muscle for injection?
o Line 222: there are no studies regarding SMNBs available in literature who can helpè there are no studies regarding SMNBs available in literature which can help
Reply to the comments of Reviewer 1
The authors would like to express their gratitude to the Reviewer for the positive feedback and useful comments. We feel that revision of the manuscript according to your suggestions has strengthened this review. We have complied with each of the issues raised and have explained the revision in a point-by-point fashion (presented below) in italic.
Reply to the comments of Reviewer 1:
Minor revisions:
- As suggested by the Reviewer, grammatical and spelling corrections were carried out. We thank the Reviewer for highlighting these errors.
E.g.:
- Appropriatepositioningè appropriate positioning
- In clinical practice Botulinum toxin A (BTA) is regarded asè In clinical practice, Botulinum toxin A (BTA) is regarded as
- It is reported as the third most leading cause of disability (29)è It is reported as the third most leading cause of disability
- Also, some references were not with the correct numbering or additional referencing needed to be added.
E.g.:
- Corrected in the manuscript. (29) deleted.
-concerning ref. 2. The additional reference ‘D.E. McLean. Medical complications experienced by a cohort of stroke survivors during inpatient, tertiary-level stroke rehabilitation. Arch Phys Med Rehabil, 85 (2004), pp. 466-469’ was wrongly left out at one point during revision.
- What is the Clinical manifestation of spastic hemiplegic shoulder pain? è What are the clinical manifestations of spastic hemiplegic shoulder pain?
- Due to the wide variety of possible manifestations of spasticity its assessment becomes very complex. è Due to the wide variety of possible manifestations of spasticity, its assessment becomes very complex.
- What are adequate outcome measures of spasticity? The authors should emphasize the outcome measure of “shoulder” spasticity. Are there any with particular interest for “shoulder”. The authors should also discuss adequate outcome measure of ”pain”, as they briefly discuss later the possible effect of botulinum toxin on pain.
We thank the Reviewer for highlighting this important issue. Unfortunately, there are no specific outcome measures found in literature concerning shoulder spasticity. We added this point of discussion to the manuscript. However, we added info regarding a region-specific outcome measure, the Shoulder Pain and Disability Index. Also, we added a short paragraph on the “pain” subject, and also added ‘ pain’ to the title. Spastic hemiplegic shoulder pain is in fact the scope of the article. The title covers the load of the article now.
- Line 222: there are no studies regarding SMNBs available in literature who can helpè there are no studies regarding SMNBs available in literature which can help
At this point, we feel to have addressed each issue raised by Reviewer 1
Reviewer 2 Report
1. L16-17 and 39-40 – ‘’ 150.5 per 100,000 in 2017’’ - it is not clear. E.g. it should be clarified that it is about people/diseases.
2. L41 –‘’(29)’’ - What does this mean? Is it quoting ?
3. L49 – ‘’(61)’’ - What does this mean? Is it quoting ?
4. L53 – ‘’ (>14/42(>medium score)’’ - This is a record that is difficult to understand. Please reword it.
5. L54 – ‘’ Institutes of Health Stroke Scale score) and spasticity [4]’’ - Quote #4 is incorrect here. The authors do not refer to the Institute's website or official recommendations.
6. L63 - Table 1- First, the title is missing. In my opinion, it is not the table but the graphics. And graphics of poor quality and not much contribution. I suggest to improve it in its entirety.
7. L79 - Introduction - Please add information about post-stroke physiotherapy.
8. I will remind authors that if an abbreviation appears for the first time it should be developed. And so please feel free to elaborate:
a. L152 – ‘’ BTX’’
b. L161 – ‘’EMG’’
c. 162 – ‘’F/M’’
d. L167 – ‘’RCT’’
9. L167 –‘’ As an illustration’’ - Suggests replacing the phrase.
10. L174 – ‘’ US. US’’ - I suggest swapping repetitions in succession.
11. L209- ‘’ (83)’’ - What does this mean? Is it quoting ?
12. L226-227 - In my opinion, this study is not necessary. It represents a repetition of a sentence from line 191-192.
13. L168, 236, 289 and 297 - When authors write sufrence EMG (L 161 and 166) - please write so throughout the text or sEMG. The very abbreviation EMG suggests the possibility of needle electromyography. Which electromyography are the authors referring to ?
14. L374 – In my opinion, it is not the table but the graphics. The graphics of poor quality and not much contribution. I suggest to improve it in its entirety.
15. Table 2. – ‘’610’’ - What does this mean? Is it an error?
16. 4. Discussion
a. First, the discussion is for improvement in its entirety. It only focuses in a small part on the results of the review. It should elaborate on what the authors describe in the paper.
b. L393-394 – ‘’ selection bias because of the narrative ‘’ - If there is such a risk, why didn't the authors decide to use, for example, QUADAS or QUADAS-2 tool?
c. L396 – ‘’ systematic search strategy’’ - Please describe what the authors have in mind. There is too much generality.
17. L 421 - Data Availability Statement: - incorrect form of the statement.
18. References - References does not follow the style of mdpi.
Reply to the comments of Reviewer 2
The authors would like to express their gratitude to the Reviewer for the positive feedback and useful comments. We feel that revision of the manuscript according to your suggestions has strengthened this review. We have complied with each of the issues raised and have explained the revision in a point-by-point fashion (presented below) in italic.
Reply to the comments of Reviewer 2:
As suggested by the Reviewer, grammatical and spelling corrections were carried out. We thank the Reviewer for highlighting these errors.
- L16-17 and 39-40 – ‘’ 150.5 per 100,000 in 2017’’ - it is not clear. E.g. it should be clarified that it is about people/diseases.
- We added the word ‘population’ in accordance with the article we referred to.
- L41 –‘’(29)’’ - What does this mean? Is it quoting ?
- Deleted in the manuscript. Indeed, this was a reference from an early version of the manuscript forgotten to delete.
- L49 – ‘’(61)’’ - What does this mean? Is it quoting ?
- Deleted in the manuscript. Indeed, this was a reference from an early version of the manuscript forgotten to delete.
- L53 – ‘’ (>14/42(>medium score)’’ - This is a record that is difficult to understand. Please reword it.
- As suggested by the reviewer. We clarified the score in the manuscript in line 53: ‘the severity of the disability (>14/42(corresponding to a medium severity) on the National Institutes of Health Stroke Scale score) and spasticity [4] (table 1).’
- L54 – ‘’ Institutes of Health Stroke Scale score) and spasticity [4]’’ - Quote #4 is incorrect here. The authors do not refer to the Institute's website or official recommendations.
- We don’t fully understand the remark. Quote 4 corresponds fully with the previous sentence summarizing the predictors of HSP, including the severity of the disability (>14/42(corresponding to a medium severity) on the National Institutes of Health Stroke Scale score). We would like to invite the reviewer to clarify the remark if at which point the quoting is incorrect.
- L63 - Table 1- First, the title is missing. In my opinion, it is not the table but the graphics. And graphics of poor quality and not much contribution. I suggest improving it in its entirety.
- We adjusted the title and the table in the manuscript.
- L79 - Introduction - Please add information about post-stroke physiotherapy.
- Physiotherapy post stroke is not the scope of this article, neither are orthotics, occupational therapy and other pharmacological therapies. We added the following phrases to the manuscript on possible relevant adjunct therapies: ‘It should be stated that BTA in HSP is not a stand-alone therapy. Although they are not the scope of this article, adjunct therapies like stretching and active or passive physiotherapy techniques may improve the treatment results’
- I will remind authors that if an abbreviation appears for the first time it should be developed. And so please feel free to elaborate:
- L152 – ‘’ BTX’’
- L161 – ‘’EMG’’
- 162 – ‘’F/M’’
- L167 – ‘’RCT’’
- Thanks for the attentive remarks to the reviewer. Corrections are made in the manuscript
- L167 –‘’ As an illustration’’ - Suggests replacing the phrase.
- As an illustration is replaced by ‘ in fact’ in line 188 ( numbering changed due to insertions of new content)
- L174 – ‘’ US. US’’ - I suggest swapping repetitions in succession.
- Indeed, good remark. We deleted one ‘US’ in the manuscript.
- L209- ‘’ (83)’’ - What does this mean? Is it quoting ?
- The erroneous number (83) is a quote from a previous version of the manuscript. Numbering is corrected in the manuscript.
- L226-227 - In my opinion, this study is not necessary. It represents a repetition of a sentence from line 191-192.
- Sentence ( in new numbering ) 249 which is indeed a repetition, although ment to summarize the chapter ( done in each chapter) is deleted
- L168, 236, 289 and 297 - When authors write surface EMG (L 161 and 166) - please write so throughout the text or sEMG. The very abbreviation EMG suggests the possibility of needle electromyography. Which electromyography are the authors referring to?
- As suggested by the reviewer surface- and needle- are added where necessary to EMG throughout the text in order to make it clear for the reader.
- L374 – In my opinion, it is not the table but the graphics. The graphics of poor quality and not much contribution. I suggest to improve it in its entirety.
- adjustments are made to the table/graphics
- Table 2. – ‘’610’’ - What does this mean? Is it an error?
- Indeed, the error is deleted
- 4. Discussion
- First, the discussion is for improvement in its entirety. It only focuses in a small part on the results of the review. It should elaborate on what the authors describe in the paper.
- The article is a scoping review. That’s why most of the content is already written in the previous chapters. It is therefore, unfortunately not leading to and elaborated discussion. On the other hand, the discussion doesn’t follow the chronology of the article entirely. That’s why we moved the last paragraph to the beginning of the discussion, corresponding the chronology of the article.
- L393-394 – ‘’ selection bias because of the narrative ‘’ - If there is such a risk, why didn't the authors decide to use, for example, QUADAS or QUADAS-2 tool? c. L396 – ‘’ systematic search strategy’’ - Please describe what the authors have in mind. There is too much generality.
It is correctly mentioned by the Reviewer that this point of discussion wasn’t fully clear. We have changed this paragraph; it now reads as follow:
There can be a concern about selection bias because of the narrative character of this review. However, a systematic review methodology was originally planned but an initial systematic search identified the unstandardized diverse nature of the current BTA related studies, and the small body of available relevant work made a systematic approach and meta-analysis of data not feasible. Also, the narrative approach provided us with a more nuanced and clinically applicable review to extend knowledge and set research priorities in the area.
- L 421 - Data Availability Statement: - incorrect form of the statement.
Thank you for mentioning, we have added the following statement:
Data sharing not applicable. No new data were created or analyzed in this study. Data sharing is not applicable to this article.
- References - References does not follow the style of mpdi.
The correct reference style is not clear to the authors. We have inserted the references using the software Endnote, which makes it easy for the journal to adjust according there requested style.
At this point, we feel to have addressed each issue raised by Reviewer 2